# Grounding Dialogue Systems via Knowledge Graph Aware Decoding with Pre-trained Transformers

Debanjan Chaudhuri[2] [*], Md Rashad Al Hasan Rony[1] [*], and Jens Lehmann[1,2]

[1] Fraunhofer IAIS, Dresden, Germany
`rashad.rony@iais.fraunhofer.de`, `jens.lehmann@iais.fraunhofer.de`
[2] Smart Data Analytics Group, University of Bonn
`s6dechau@uni-bonn.de`, `jens.lehmann@cs.uni-bonn.de`

**Abstract.** Generating knowledge grounded responses in both goal and non-goal oriented dialogue systems is an important research challenge. Knowledge Graphs (KG) can be viewed as an abstraction of the real world, which can potentially facilitate a dialogue system to produce knowledge grounded responses. However, integrating KGs into the dialogue generation process in an end-to-end manner is a non-trivial task. This paper proposes a novel architecture for integrating KGs into the response generation process by training a BERT model that learns to answer using the elements of the KG (entities and relations) in a multi-task, end-to-end setting. The k-hop subgraph of the KG is incorporated into the model during training and inference using Graph Laplacian. Empirical evaluation suggests that the model achieves better knowledge groundedness (measured via Entity F1 score) compared to other state-of-the-art models for both goal and non-goal oriented dialogues.

**Keywords:** Knowledge graph · Dialogue system · Graph encoding · Knowledge integration.

## 1 Introduction

Recently, dialogue systems based on KGs have become increasingly popular because of their wide range of applications from hotel bookings, customer-care to voice assistant services. Such dialogue systems can be realized using both goal and non-goal oriented methods. Whereas the former one is employed for carrying out a particular task, the latter is focused on performing natural ("chit-chat") dialogues. Both types of dialogue system can be implemented using a generative approach. In a generative dialogue system, the response is generated (usually word by word) from the domain vocabulary given a natural language user query, along with the previous dialogue context. Such systems can benefit from the integration of additional world knowledge [12]. In particular, knowledge graphs, which are an abstraction of real world knowledge, have been shown to be useful for this purpose. Information of the real world can be stored in a KG in a structured (Resource Description Framework (RDF) triple, e.g., $< subject, relation, object >$) and abstract way (Paris is the capital city of France and be presented in $< Paris, capital\ city, France >$). KG based question answering

---

[*] Equal contribution

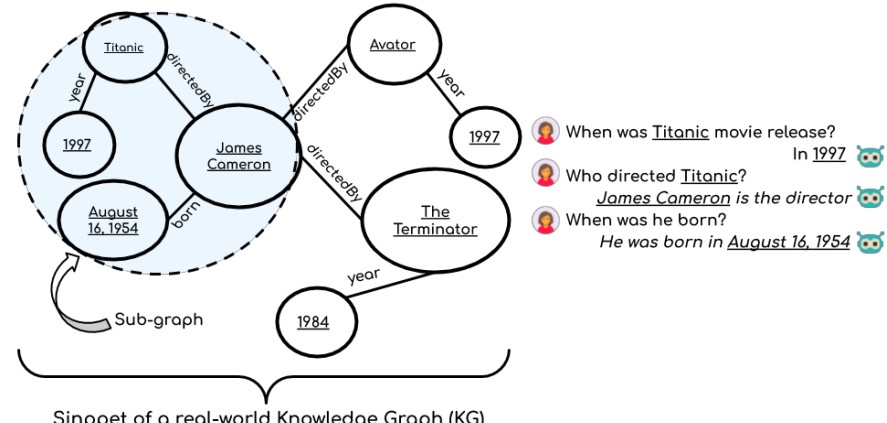

Fig. 1: Example of a knowledge grounded conversation.

(KGQA) is already a well-researched topic [6]. However, generative dialogue systems with integrated KGs have only been explored more recently [7, 20, 12]. To model the response using the KG, all current methods assume that the entity in the input query or a sub-graph of the whole KG, which can be used to generate the answer, is already known [20, 36]. This assumption makes it difficult to scale such systems to real-world scenarios, because the task of extracting sub-graphs or, alternatively, performing entity linking in large knowledge graphs is non-trivial [27]. An example of a knowledge graph based dialogue system is shown in Figure 1. In order to generate the response *James Cameron is the director*, the system has to link the entity mentioned in the question in the first turn i.e. *Titanic*, and identify the relation in the KG connecting the entities *Titanic* with *James Cameron*, namely *directed by*. Additionally, to obtain a natural dialogue system, it should also reply with coherent responses (eg. "James Cameron is the director") and should be able to handle small-talk such as greetings, humour etc. Furthermore, in order to perform multi-turn dialogues, the system should also be able to perform co-reference resolution and connect the pronoun (*he*) in the second question with *James Cameron*.

In order to tackle these research challenges, we model the dialogue generation process by jointly learning the entity and relation information during the dialogue generation process using a pre-trained BERT model in an end-to-end manner. The model's response generation is designed to learn to predict relation(s) from the input KG instead of the actual object(s) (intermediate representation). Additionally, a graph Laplacian based method is used to encode the input sub-graph and use it for the final decoding process.

Experimental results suggest that the proposed method improves upon previous state-of-the-art approaches for both goal and non-goal oriented dialogues. Our code is publicly available on Github [3]. Overall, the contributions of this paper are as follows:

---

[3]https://github.com/SmartDataAnalytics/kgirnet/

– A novel approach, leveraging the knowledge graph elements (entities and relations) in the questions along with pre-trained transformers, which helps in generating suitable knowledge grounded responses.
– We have also additionally encoded the sub-graph structure of the entity of the input query with a Graph Laplacian, which is traditionally used in graph neural networks. This novel decoding method further improves performance.
– An extensive evaluation and ablation study of the proposed model on two datasets requiring grounded KG knowledge: an in-car dialogue dataset and soccer dialogues for goal and non-goal oriented setting, respectively. Evaluation results show that the proposed model produces improved knowledge grounded responses compared to other state-of-the-art dialogue systems w.r.t. automated metrics, and human-evaluation for both goal and non-goal oriented dialogues.

## 2    Model Description

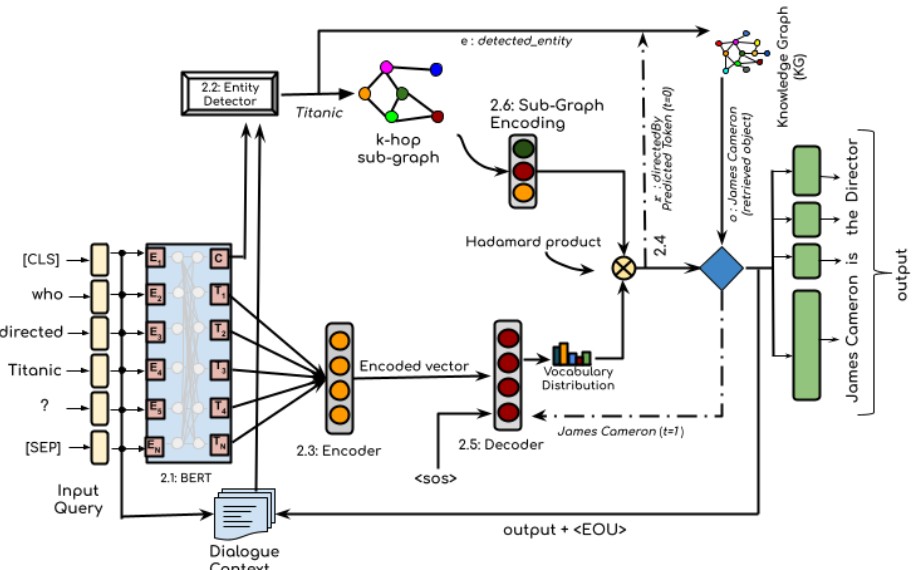

Fig. 2: KGIRNet model diagram.

We aim to solve the problem of answer generation in a dialogue using a KG as defined below.

**Definition 1  (Knowledge Graph).** *Within the scope of this paper, we define a* knowledge graph $KG$ *as a labelled, undirected multi-graph consisting of a set $V$ of nodes and a set $E$ of edges between them. There exists a function, $f_l$ that maps the nodes and*

*vertices of a graph to a string. The neighborhood of a node of radius $k$ (or $k$-hop) is the set of nodes at a distance equal to or less than $k$.*

This definition is sufficiently generic to be applicable to knowledge graphs based on RDF[4] (Resource Description Framework) as well as property graphs [13]. The vertices $V$ of the $KG$ represent entities $e \in V$, while the edges represent the relationships between those entities. A fact is an ordered triple consisting of an entity $e$ (or subject $s$), an object $o$ and the relation $r$ (a.k.a. predicate $p$) between them, denoted by $(s, p, o)$.

The proposed model for dialogue generation, which we call KGIRNet, is quintessentially a sequence-to-sequence based model with a pre-trained transformer serving as its input as illustrated in Figure 2. In contrast to previous works, we introduce an intermediate query representation using the relation information, for training. We also employ a Graph Laplacian based method for encoding the input sub-graph of the KG that aids in predicting the correct relation(s) as well as filter out irrelevant KG elements. Our approach consists of the following steps for which more details are provided below: Firstly, we encode the input query $q$ with a pre-trained BERT model (Section 2.1). Next we detect a core entity $e$ occurring in $q$ (Section 2.2). Input query $q$ and generated output is appended to the dialogue context in every utterance. The input query, encoded using the BERT model is then passed through an LSTM encoder to get an encoded representation (Section 2.3). This encoded representation is passed onto another LSTM decoder (Section 2.5), which outputs a probability distribution for the output tokens at every time-step. Additionally, the model also extracts the $k$-hop neighbourhood of $e$ in the KG and encodes it using graph based encoding (Section 2.6) and perform a Hadamard product with token probability distribution from the decoder. The decoding process stops when it encounters a special token, $< EOS >$ (end of sentence). Dotted lines in the model diagram represent operations performed at a different time-step $t$ in the decoding process where solid lines are performed once for each utterance or input query.

In this work, we define complex questions as questions which require multiple relations to answer the given question. For example, for the following query: "*please tell me the location, time and the parties that are attending my meeting*", the model needs to use 3 relations from the KG for answering, namely location, time and parties. The answer given by the model could be : "*you have meeting scheduled on friday at 10am with boss in conference_room_102 to go_over_budget*". The model is able to retrieve important relation information from the KG during decoding. However, the model is not able to handle questions which go beyond the usage of explicitly stored relations and require inference capabilities .

### 2.1   Query Encoding

BERT is a pre-trained multi-layer, bi-directional transformer [32] model proposed in [8]. It is trained on unlabelled data for two-phased objective: masked language model and next sentence prediction. For encoding any text, special tokens [CLS] and [SEP] are inserted at the beginning and the end of the text, respectively. In the case of KGIRNet,

---

[4]https://www.w3.org/RDF/

the input query $q = (q_1, q_2, ...q_n)$ at turn $t_d$ in the dialogue, along with the context up to turn $t_d - 1$ is first encoded using this pre-trained BERT model which produces hidden states $(T_1, T_2....T_n)$ for each token and an aggregated hidden state representation $C$ for the [CLS] (first) token. We encode the whole query $q$ along with the context, concatenated with a special token, $< EOU >$ (end of utterance).

## 2.2  Entity Detection

The aggregated hidden representation from the BERT model $C$ is passed to a fully connected hidden layer to predict the entity $e_{inp} \in V$ in the input question as given by

$$e_{inp} = softmax(w_{ent}C + b_{ent}) \qquad (1)$$

Where, $w_{ent}$ and $b_{ent}$ are the parameters of the fully connected hidden layer.

## 2.3  Input Query Encoder

The hidden state representations $(T_1, T_2....T_n)$ of the input query $q$ (and dialogue context) using BERT is further encoded using an LSTM [14] encoder which produces a final hidden state at the $n$-th time-step given by

$$h_n^e = f_{enc}(T_n, h_{n-1}^e) \qquad (2)$$

$f_{enc}$ is a recurrent function and $T_n$ is the hidden state for the input token $q_n$ from BERT. The final representation of the encoder response is a representation at every $n$ denoted by

$$H_e = (h_0^e, h_1^e....h_N^e) \qquad (3)$$

## 2.4  Intermediate Representation

As an intermediate response, we let the model learn the relation or edge label(s) required to answer the question, instead of the actual object label(s). In order to do this, we additionally incorporated the relation labels obtained by applying the label function $f_l$ to all edges in the KG into the output vocabulary set. If the output vocabulary size for a vanilla sequence-to-sequence model is $v_o$, the total output vocabulary size becomes $v_{od}$ which is the sum of $v_o$ and $v_{kg}$. The latter being the labels from applying the $f_l$ to all edges (or relations) in the KG.

For example, if in a certain response, a token corresponds to an object label $o_l$ (obtained by applying $f_l$ to $o$) in the fact $(e, r, o)$, the token is replaced with a KG token $v_{kg}$ corresponding to the edge or relation label $r_l$ of $r \in E$ in the KG. During training, the decoder would see the string obtained by applying $f_l$ to the edge between the entities *Titanic* and *James Cameron*, denoted here as *r:directedBy*. Hence, it will try to learn the relation instead of the actual object. This makes the system more generic and KG aware, and easily scalable to new facts and domains.

During evaluation, when the decoder generates a token from $v_{kg}$, a KG lookup is done to decode the label $o_l$ of the node $o \in V$ in the KG ($V$ being the set of nodes or vertices in the KG). This is generally done using a SPARQL query.

## 2.5   Decoding Process

The decoding process generates an output token at every time-step $t$ in the response generation process. It gets as input the encoded response $H_e$ and also the KG distribution from the graph encoding process as explained later. The decoder is also a LSTM, which is initialized with the encoder last hidden states and the first token used as input to it is a special token, $< SOS >$ (start of sentence). The decoder hidden states are similar to that of the encoder as given by the recurrent function $f_{dec}$

$$h_n^d = f_{dec}(w_{dec}, h_{n-1}^d) \tag{4}$$

This hidden state is used to compute an attention over all the hidden states of the encoder following [19], as given by

$$\alpha_t = softmax(W_s(tanh(W_c[H_e; h_t^d])) \tag{5}$$

Where, $W_c$ and $W_s$ are the weights of the attention model. The final weighted context representation is given by

$$\tilde{h}_t = \sum_t \alpha_t h_t \tag{6}$$

This representation is concatenated (represented by ;) with the hidden states of the decoder to generate an output from the vocabulary with size $v_{od}$.
The output vocab distribution from the decoder is given by

$$O_{dec} = W_o([h_t; \tilde{h_t^d}]) \tag{7}$$

In the above equation, $W_o$ are the output weights with dimension $\mathbf{R}^{h_{dim} X v_{od}}$. $h_{dim}$ being the dimension of the hidden layer of the decoder LSTM. The total loss is the sum of the vocabulary loss and the entity detection loss. Finally, we use beam-search [31] during the the decoding method.

## 2.6   Sub-Graph Encoding

In order to limit the KGIRNet model to predict only from those relations which are connected to the input entity predicted from step 2.2, we encode the sub-graph along with its labels and use it in the final decoding process while evaluating.
The $k$-hop sub-graph of the input entity is encoded using Graph Laplacian [16] given by

$$G_{enc} = D^{-1}\tilde{A}f_{in} \tag{8}$$

Where, $\tilde{A} = A + I$. $A$ being the adjacency matrix, $I$ is the identity matrix and D is the degree matrix. $f_{in}$ is a feature representation of the vertices and edges in the input graph. $G_{enc}$ is a vector with dimensions $\mathbf{R}^{ik}$ corresponding to the total number of nodes and edges in the $k$-hop sub-graph of $e$. An example of the sub-graph encoding mechanism is shown in Figure 3.
The final vocabulary distribution $O_f \in R^{v_{od}}$ is a Hadamard product of this graph vector and the vocabulary distribution output from the decoder.

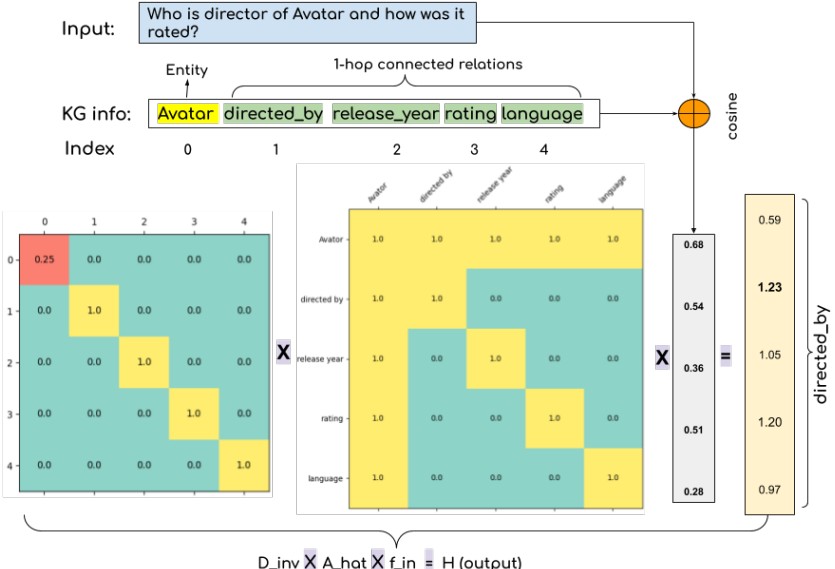

Fig. 3: Sub-Graph Encoding using Graph Laplacian.

$$O_f = O_{dec} \odot G_{enc} \tag{9}$$

This step essentially helps the model to give additional importance to the relations connected at k-hop based on its similarity with the query also to filter(mask) out relations from the response which are not connected to it. For the query in 3 *who is the director of Avatar* and how was it rated ? The graph laplacian based encoding method using only relation labels already gives higher scores for the relations directed_by and rating, which are required by the question. This vector when multiplied with the final output vocabulary helps in better relation learning.

## 3    Experimental Setup

### 3.1    Datasets

Available datasets for knowledge grounded conversations are the *in-car dialogue* data as proposed by [12] and *soccer dialogues* over football club and national teams using a knowledge graph [7]. The former contains dialogues for assisting the driver of a car with data related to weather, schedules and navigation, in a goal-oriented setting. The soccer dataset contains non-goal oriented dialogues about questions on different soccer teams along with a knowledge graph consisting of facts extracted from Wikipedia about the respective clubs and national soccer teams. Both the datasets were collected using Amazon Mechanical Turk (AMT) by the respective authors [12, 7]. The statistics of the datasets are provided in Table 1. As observed, the number of dialogues for both the dataset is low.

To perform a KG grounded dialogue as in KG based question-answering [10], it is important to annotate the dialogues with KG information such as the entities and relations, the dialogues are about. Such information were missing in the soccer dataset, hence we have semi-automatically annotated them with the input entity $e_{inp}$ in the query $q$ and the relations in the $k$-hop sub-graph of the input entity required to answer $q$. For the domain of in-car it was possible to automatically extract the entity and relation information from the dialogues and input local KG snippets.

Table 1: Dataset statistics.

|  | In-car dialogues | | | Soccer dialogues | | |
|---|---|---|---|---|---|---|
| Number of triples, entity, relations | 8561, 271, 36 | | | 4301, 932, 30 | | |
|  | Train | Validation | Test | Train | Validation | Test |
| Number of dialogues | 2011 | 242 | 256 | 1328 | 149 | 348 |
| Number of utterances | 5528 | 657 | 709 | 6523 | 737 | 1727 |
| KG-grounded questions (%) | 44.95 | 33.94 | 43.84 | 6.53 | 4.61 | 3.88 |

### 3.2   Evaluation Metrics

We evaluate the models using the standard evaluation metrics BLEU [25] and Entity F1 scores as used in the discussed state-of-the-art models. However, unlike [20] we use Entity F1 scores based on the nodes $V$ of the KG, as inspired by previous works on KG based question answering [3]. Additionally, we use METEOR [1] because it correlates the most with human judgement scores [30].

### 3.3   Model Settings

Table 2: Evaluation on goal and non-goal oriented dialogues.

| Models | In-Car Dialogues | | | Soccer Dialogues | | | Inference time |
|---|---|---|---|---|---|---|---|
|  | BLEU | Entity F1 | METEOR | BLEU | Entity F1 | METEOR | utterances/sec |
| Seq2Seq | 4.96 | 10.70 | 21.20 | 1.37 | 0.0 | 7.8 | 133 |
| Mem2Seq[20] | 9.43 | 24.60 | 28.80 | 0.82 | 04.95 | 7.8 | 88 |
| GLMP[36] | 9.15 | 21.28 | 29.10 | 0.43 | 22.40 | 7.7 | 136 |
| Transformer[32] | 8.27 | 24.79 | 29.06 | 0.45 | 0.0 | 6.7 | 7 |
| DialoGPT[39] | 7.35 | 21.36 | 20.50 | 0.76 | 0.0 | 5.5 | 2 |
| KG-Copy[7] | - | - | - | **1.93** | 03.17 | **10.89** | 262 |
| KGIRNet | **11.76** | **32.67** | **30.02** | 1.51 | **34.33** | 8.24 | 37 |

For entity detection we used a fully connected layer on top of CNN-based architecture. Size of the hidden layer in the fully connected part is 500 and a dropout value of 0.1 is used and ReLU as the activation function. In the CNN part we have used 300 filters

with kernel size 3, 4 and 5. We use BERT-base-uncased model for encoding the input query. The encoder and decoder is modelled using an LSTM (long short term memory) with a hidden size of 256 and the KGIRNet model is trained with a batch size of 20. The learning rate of the used encoder is 1e-4. For the decoder we select a learning rate of 1e-3. We use the Adam optimizer to optimize the weights of the neural networks. For all of our experiments we use a sub-graph size of k=2. For calculating $f_{in}$ as in Equation 8, we use averaged word embedding similarity values between the query and the labels of the sub-graph elements. The similarity function used in this case is cosine similarity. We save the model with best validation Entity f1 score.

Table 3: Relation Linking Accuracy on SQB [37] Dataset.

| Method | Model | Accuracy |
|--------|-------|----------|
| **Supervised** | Bi-LSTM [21] | 38.5 |
| | HR-LSTM [38] | 64.3 |
| **Unsupervised** | Embedding Similarity | 60.1 |
| | Graph Laplacian (this work) | **69.7** |

## 4   Results

In this section, we summarize the results from our experiments. We evaluate our proposed model KGIRNet against the current state-of-the-art systems for KG based dialogues; namely Mem2Seq [20], GLMP [36], and KG-Copy [7][5]. To include a baseline method, the results from a vanilla Seq2Seq model using an LSTM encoder-decoder are also reported in Table 2, along with a vanilla transformer [32] and a pre-trained GPT-2 model, DialoGPT [39] fine-tuned individually on both the datasets.

We observe that KGIRNet outperforms other approaches for both goal (in-car) and non-goal oriented (soccer) dialogues except for BLEU and METEOR scores on the soccer dataset. The effect of knowledge groundedness is particularly visible in the case of soccer dialogues, where most models (except GLMP) produces feeble knowledge grounded responses. The Entity F1 score used here is adapted from [3] and is defined as the average of F1-scores of the set of predicted objects, for all the questions in the test set.

In addition to evaluating dialogues, we also evaluate the proposed graph laplacian based relation learning module for the task of knowldge-graph based relation linking. Although, it is a well-researched topic and some systems claim to have solved the problem [26], but such systems are not able to handle relations which are not present in the training data [37]. The latter also proposed a new, balanced dataset (SQB) for simple question answering which has same proportions of seen and unseen relations in the test or evaluation set. We have evaluated our unsupervised graph laplacian based method for relation linking on the SQB dataset against supervised methods namely Bi-LSTM [21], hr-bilstm [38] and unsupervised method such as text embedding based similarity between the query and the relations connected to the subject entity with 1-hop.

---

[5]KG-Copy reports on a subset of the in-car testset hence it is not reported here

The results are reported in Table 3. As observed, Graph Laplacian performs better wrt. supervised methods on unseen relations and also better than shallow embedding based similarity. This is one of the motivation for using this simple method during KGIRNet's decoding process. We run all the inference on a setup with 32GB of RAM and 8GB of VGA and a Bandwidth of 256.3 GB/s.

## 5    Discussion

For in-car dialogues, we train and evaluate on queries which require knowledge from the KG, hence we omit the scheduling conversations where the driver asks the car to schedule a meeting/conference. In order to get the knowledge graph candidate triples for all the other models (Mem2Seq and GLMP), we provide them with the 2-hop sub-graph of the correct input entity instead of the local knowledge as used in the respective models; this, represents a scenario closer to a real-world KG grounded conversation. For the in-car dataset, the human baseline BLEU score as reported in [12] is 13.5 (the KGIRNet score is 11.76). The BLEU scores for soccer are low because non-goal oriented dialogues are more complicated to model due to large vocabulary sizes (more than 3 times the vocabulary size of the of in-car dataset). Also in the case of soccer dialogues, number of factual conversation is low (4%) compared to the total number of utterances and also the conversations are more generic and repetitive in nature.

Table 4: Analyzing sample predictions.

| Input Query | True Response | Predicted Responses | | |
|---|---|---|---|---|
| | | **GLMP** | **Mem2Seq** | **KGIRNet** |
| **(S)** who is currently coaching bvb dortmund | lucien favre | *the is the coach* | *yes , they have a good* | *lucien favre is the coach of bvb dortmund* |
| **(S)** when did italy last win the world cup | 2006 | *italy won the world cup in 2006* | *i think they have a good team* | *italy won the world cup in 2006* |
| **(C)** what time is my doctorappointment? | your doctorappointment is scheduled for friday at 11am | *your next is is at 1pm at 7pm* | *your doctorappointment is at 1pm* | *your doctorappointment is on friday at 11am* |
| **(C)** i need gas | valero is 4_miles away | *there is a valero away* | *chevron is gas_station away chevron is at away* | *there is a valero nearby* |

### 5.1    Human Evaluation

We perform a human-based evaluation on the whole test dataset of the generated responses from the KGIRNet model and its closest competitors, i.e. Mem2Seq, GLMP and DialoGPT models. We asked 3 internal independent annotators who are not the authors of this paper (1 from CS and 2 from non-CS background) to rate the quality of the generated responses between 1-5 with respect to the dialogue context (higher is better). Note that for each dataset, we provide the annotators with 4 output files in CSV format (containing the predictions of each model) and the knowledge graph in RDF format. Each of the CSV files contains data in 5 columns: question, original response, predicted response, grammatical correctness, similarity between original and predicted response. In the provided files, the 4th (grammatically correctness) and 5th (similarity between

original and predicted response) columns are empty and we ask the annotators to fill them with values (within a range of 1-5) according to their judgement. The measures requested to evaluate upon are correctness (Corr.) and human-like (human) answer generation capabilities. Correctness is measured by comparing the response to the correct answer object. Reference responses are provided besides the system generated response in order to check the factual questions. The results are reported in Table 5a. Cohen's Kappa of the annotations is 0.55.

## 5.2   Ablation Study

As an ablation study we train a sequence-to-sequence model with pre-trained fasttext embeddings as input (S2S), and the same model with pre-trained BERT as input embedding (S2S_BERT). Both these models do not have any information about the structure of the underlying knowledge graph. Secondly, we try to understand how much the intermediate representation aids to the model, so we train a model (KGIRNet_NB) with fasttext embeddings as input instead of BERT along with intermediate relation representation. Thirdly, we train a model with pre-trained BERT but without the haddamard product of the encoded sub-graph and the final output vocabulary distribution from Step 2.6. This model is denoted as KGIRNet_NS. As observed, models that are devoid of any KG structure has very low Entity F1 scores, which is more observable in the case of soccer dialogues since the knowledge grounded queries are very low, so the model is not able to learn any fact(s) from the dataset. The proposed intermediate relation learning along with Graph Laplacian based sub-graph encoding technique observably assists in producing better knowledge grounded responses in both the domains; although, the latter aids in more knowledge groundedness in the domain of soccer dialogues (higher Entity F1 scores). We also did an ablation study on the entity detection accuracy of the end-to-end KGIRNet model in the domain of in-car and compared it with a standalone Convolutional neural network (CNN) model which predicts the entity from the input query, the accuracies are 79.69 % and 77.29% respectively.

Table 5: In-depth evaluation of KGIRNet model.

| Models | In-Car | | Soccer | |
|---|---|---|---|---|
| | Corr. | Human | Corr. | Human |
| Mem2Seq | 3.09 | 3.70 | 1.14 | 3.48 |
| GLMP | 3.01 | 3.88 | 1.10 | 2.17 |
| DialoGPT | 2.32 | 3.43 | 1.32 | 3.88 |
| KGIRNet | 3.60 | 4.42 | 1.59 | 3.78 |

(a) Human evaluation.

| Models | In-Car Dialogues | | Soccer Dialogues | |
|---|---|---|---|---|
| | BLEU | EntityF1 | BLEU | EntityF1 |
| S2S | 4.96 | 10.70 | 1.49 | 0.0 |
| S2S_BERT | 7.53 | 09.10 | 1.44 | 0.0 |
| KGIRNet_NB | 9.52 | 29.03 | 0.91 | 29.85 |
| KGIRNet_NS | 11.40 | 33.03 | 1.05 | 28.35 |
| KGIRNet | 11.76 | 32.67 | 1.51 | 34.32 |

(b) Ablation study.

## 5.3   Qualitative Analysis

The responses generated by KGIRNet are analyzed in this section. Responses from some of the discussed models along with the input query are provided in Table 4 [6]. We

---

[6]In the table **(S)** and **(C)** refers to example from Soccer and In-car dataset respectively.

compare the results with two other state-of-the-art models with the closest evaluation scores, namely Mem2Seq and GLMP. The first two responses are for soccer dialogues, while the latter two are for in-car setting. We inspect that qualitatively the proposed KGIRNet model produces better knowledge grounded and coherent responses in both the settings. In the case of soccer dialogues, predicting single relation in the response is sufficient, while for the case of in-car dialogues, responses can require multiple relation identification. KGIRNet model is able to handle such multiple relations as well (e.g., $r$:date friday and $r$:time 11am for the third utterance).

Table 6: Analyzing fact-fullness of KGIRNet.

| Input Query | True Response | Predicted Response | Intermediate Response |
|---|---|---|---|
| who is senegal captain ? | cheikhou kouyate | *sadio mane is the captain of senegal* | *r:captain is the captain of @entity* |
| who is the goalie for iceland | hannes halldorsson | *runar alex runarsson* | *r goalkeeper* |

To understand more about the advantages of KG grounded generation process, consider the examples from the soccer dataset below in Table 6, in which the model has failed. We have also included the intermediate responses that are generated from the decoder before the KG lookup phase for the reference. In the first case, while constructing the KG, the captain was *Sadio Mane*, whereas the during the dialogue collection phase the probable captain was *Checikhou Kouyate* (as per the annotator). These errors on temporal information can easily be tackled using Knowledge Grounded dialogue systems, where the system can potentially reflect current information with updated real world knowledge. For the second case, the current main goalkeeper for the Iceland national football team is *Hannes Halldórsson*; however, *Rúnar Alex Rúnarsson* is also among the 3 goalkeepers for the team and the model has no additional information to disambiguate between them to produce the same response as the human annotator. KGIRNet is also able to infer conversational context implicitly in dialogues with the system (Table 7).

Table 7: A conversation with KGIRNet.

| | |
|---|---|
| **User:** | what s the weather forecast for today and tomorrow ? |
| **KGIRNet:** | what city do you want the weather for |
| **User:** | los angeles |
| **KGIRNet:** | it will be 40f - 60f on monday warm on tuesday windy on wednesday hot on thursday |
| **User:** | thanks |
| **KGIRNet:** | you are welcome |

## 5.4   Error Analysis

To analyze more on the shortcomings of our proposed KGIRNet, we stated some sample erroneous responses from the model in Table 8. The first three are from soccer and the rest in-car dialogues. The model often produces generic responses for soccer as in the first example. Our model is only able to handle simple factoid queries and not able to

answer queries requiring comparative reasoning such as *best player*, *youngest player* etc, as observable from the second and third cases. In the second case, the model answers with the coach's name and when asked for the best player for Argentina and it replies with the name of an Uruguayan player in the third case. The model also produces generic relations like coach name, number of world cup championship as response when any such keyword is present in the input. For the in-car dataset, the replies are less generic and more coherent; however, in cases like the fourth example, the model is not able to perform logical reasoning (negation). Moreover, the model fails in cases where the input entity is not recognized correctly as in case of the last example. Although, the name of the friend is fetched from the vocabulary, but the predicted entity is wrong hence the model is not able to decode the relation $r$:address from the KG during decoding.  In

Table 8: Error analysis of KGIRNet.

| Input Query | True Response | Predicted Response |
|---|---|---|
| portugal win the last eurocup ? | yes they won the last europcup | *portugal has won the world cup.* |
| who is switzerlandś best player ? | xherdan shaqiri | *petkovic is the coach of switzerland* |
| who is considered their best player ? | traditionally their top scorers. | *luis suarez is the best player in argentina* |
| will it be warm in grand_rapids on saturday | it will not be warm in grand_rapids on saturday | *it is not predicted to fall on tuesday in grand_rapids* |
| where does my friend live ? | tom lives at 580_van_ness_ave | *tom s house is located at r:address* |

general, the model's entity detection fails in case of conversations with long dialogue contexts. Incorporating KG information into the entity detection process, where we consider the sub-graph of the entity in the first interaction can potentially help in further performance improvements.

## 6   Related Work

Leveraging background information for dialogue system improvement is a well-researched topic, especially in goal-oriented setting [2, 9, 35]. [12] proposed the in-car dataset which uses a knowledge base for in-car conversation about weather, location etc. Recently, [20] proposed memory-network based encoder-decoder architecture for integrating knowledge into the dialogue generation process on this dataset. Improved models in this task are proposed by [15, 36]. [7] proposed a soccer dialogue dataset along with a KG-Copy mechanism for non-goal oriented dialogues which are KG-integrated. In a slightly similar research line, in past years, we also notice the use of variational autoencoders (VAE) [40, 17] and generative adversarial networks (GANs) [23, 18] in dialogue generation. However, knowledge graph based dialogue generation is not well-explored in these approaches.

More recently, transformer-based [32] pre-trained models have achieved success in solving various downstream tasks in the field of NLP such as question answering [29] [8], machine translation [34], summarization [11]. Following the trend, a hierarchical transformer is proposed by [28] for task-specific dialogues. The authors experimented on MultiWOZ dataset [5], where the belief states are not available. However, they found

the use of hierarchy based transformer models effective in capturing the context and dependencies in task-specific dialogue settings. In a different work, [24] experimented transformer-based model on both the task-specific and non-task specific dialogues in multi-turn setting. In a recent work,  [39] investigated on transformer-based model for non-goal oriented dialogue generation in single-turn dialogue setting. Observing the success of transformer-based models over the recurrent models in this paper we also employ BERT in the dialogue generation process which improves the quality of generated dialogues (discussed in section 4 and 5).

In past years, there is a lot of focus on encoding graph structure using neural networks, a.k.a. Graph Neural Networks (GNNs) [4, 33]. In the field of computer vision, Convolutional Neural Networks (CNNs) are used to extract the most meaningful information from grid-like data structures such as images. A generalization of CNN to graph domain, Graph Convolutional Networks (GCNs) [16] has become popular in the past years. Such architectures are also adapted for encoding and extracting information from knowledge graphs [22]. Following a similar research line, in this paper, we leverage the concept of Graph Laplacian [16] for encoding sub-graph information into the learning mechanism.

## 7   Conclusion and Future Work

In this paper, we have studied the task of generating knowledge grounded dialogues. We bridged the gap between two well-researched topics, namely knowledge grounded question answering and end-to-end dialogue generation. We propose a novel decoding method which leverages pre-trained transformers, KG structure and Graph Laplacian based encoding during the response generation process. Our evaluation shows that out proposed model produces better knowledge grounded response compared to other state-of-the-art approaches, for both the task and non-task oriented dialogues.

As future work, we would like to focus on models with better understanding of text in order to perform better KG based reasoning. We also aim to incorporate additional KG structure information in the entity detection method. Further, a better handling of infrequent relations seen during training may be beneficial.

## Acknowledgement

We acknowledge the support of the excellence clusters ScaDS.AI (BmBF IS18026A-F), ML2R (BmBF FKZ 01 15 18038 A/B/C), TAILOR (EU GA 952215) and the projects SPEAKER (BMWi FKZ 01MK20011A) and JOSEPH (Fraunhofer Zukunftsstiftung).

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
