# OpenReview forum: "Grounding Dialogue Systems via Knowledge Graph Aware Decoding with Pre-trained Transformers"
_eswc-conferences.org/ESWC/2021/Conference/Research_Track — ESWC 2021 Research_

### Official Review · AnonReviewer1 · 2021-01-12
**Interesting architecture for Q&A over KGs in a multi-turn dialog setting. Confusing comparison with existing systems the paper compares its proposed approach with.**

**Confidence:** 4
**Impact:** 3
**Design And Technical Quality:** 3

**Review:**

--- I acknowledge and thank the authors for their comments and clarifications. If the accepted, authors should address some of these issues in the final version. I am still not sure what is the fraction of errors in that exist in the soccer dataset? Can "negligible" be quantified? If accepted and this issue is not addressed, the dataset with unknown amount of errors will keep propagating forward.---

This paper addresses an important challenge of incorporating external knowledge into generative dialog systems. As compared to previous works, the system proposed in this paper includes the additional task for entity detection and relation prediction.

The technical approach section is well-written and for large parts easy to follow. Going by the entity F1 scores, there is perhaps an opportunity to improve the entity detection module. While its fairly common to use a fully connected layer on top the BERT's aggregated output (for classification tasks), an off-the-shelf (or one trained over entities in dialog datasets)  named entity recognition system (NER) could give a better performance. A Bi-LSTM CRF architecture seems to be most popular one for NERs. I am also curious about why the output of BERT had to reencoded into an LSTM encoder.

Encoding the KG subgraph with Graph Laplacian and then leveraging Hadamard product between the graph vector and probability distribution of the output vocabulary is a neat trick for relation prediction for the question/query. This seems to have produced promising results.

Overall, I feel the architecture proposed in the paper is probably better suited for a multi-turn question answering dialog system (Fig. 1 from the paper) as opposed to general "chit-chat" ones. Approaches like DialoGPT and TransferTransfo [Ref1] are better suited for the latter. I am not sure this system will work well in a non-goal oriented dialog setting with minimal needs for Q&A over a knowledge graph (as conveyed by the results over the soccer dialog dataset).

I do have some concerns with regards to evaluation, not per se the raw numbers but the approach.

The two closest previous works the current paper could be potentially compared against are Mem2Seq and GLMP. The evaluation numbers reported for the in-car dataset in the Mem2Seq paper and the ones in table 2 differ (the best performing setting in the original paper report a BLEU 12.6, Entity F1 33.4). Is it because you either modified the dataset or the original systems before running the experiments for this paper? If yes, what were those changes? One such change seems to be buried in section 5 (this information should be right up front in section 4), wherein the nature of local knowledge used in the original system was changed. If the information conveyed in the 2-hop KG subgraph differs from the original local knowledge expected (e.g. 2 hop gives more/irrelevant knowledge), then probably the comparison is not fair.

 The soccer dialogues dataset seems to have errors. For instance, at line 106 in [Ref2], during the automatic entity and relation annotation process "predominately" got mapped to "p@objectominately". It looks like "red" got replaced by "@object" but in the wrong place. While the expectation is Q&A in the dialog, it's not always the case. Lines 47, 48, 69, 70 in [Ref2] and 49, 50, 95, 96 in [Ref3] demonstrate that Q&A have either flipped (responder is asking questions) or there is disconnected dialog. I had randomly picked these files to better under the data. It makes me wonder to what degree such error exists and how much it maybe impacting the overall results.

A few questions worth clarifying:
How is "correctness" in section 5 evaluated on a scale of 1-5? When the users were asked to rate on a scale of 1-5 what guidance were they given - what does 1 signify and what does 5 signify?

How was correctness measured by users for factual/KG based questions? Were they given access to the KG for each dialog?

What process was followed to annotate the dialogues with entities and relations? It seems to be leading to some errors in the soccer dataset.

What is the difference between how you measure entity F1 score v/s [18]?

Typos:

section 2.4, second paragraph: between the entities Titanic and James Cameroon --> Cameron

section 4, third paragraph: The later (--> latter) also proposed

section 4, third paragraph: The results are reported in 3 (--> in Table 3)

section 5.2, second paragraph: consider the examples from the soccer dataset below (the table is above this text) in Table 6

section 6, second paragraph: we also employ BERT in the dialogue generation process which improves the quality of generated dislodges (--> dialogues)


[Ref1] Wolf, Thomas, et al. "Transfertransfo: A transfer learning approach for neural network based conversational agents." arXiv preprint arXiv:1901.08149 (2019).

[Ref2] https://github.com/DeepInEvil/kgirnet/blob/master/data/soccer/manually_annotated/val_sketch/3087LXLJ6NSFTQPEZR2KCP0MHQIF0A.json

[Ref3] https://github.com/DeepInEvil/kgirnet/blob/master/data/soccer/manually_annotated/train_sketch/302OLP89D0J4RR3M0R9CTPRWTZ1ACK.json

**Anonymity:**

Yes, I would like my review to remain anonymous.

**Rating:**

-1: Weak Reject

**Reuse And Availability:**

4: High

**Strong Points:**

Leveraging Hadamard product between a KG subgraph vector and probability distribution of the output vocabulary produced by the input query decoder is a neat idea for relation prediction for Q&A over knowledge graphs. The overall architecture in the paper could be independently leveraged in a Q&A setting minus the dialog systems and compared against the state-of-the-art KG Q&A systems.

**Subreviewer:**

I submitted this review.

**Weak Points:**

Experimental settings for comparing the proposed architecture with Mem2Seq and GLMP seem to be unclear. The soccer dialog datasets has potentially some noise/errors. Depending on the scale of it, it may impact the results.

---

> ### Author Rebuttal · Authors · 2021-01-29
>
> Thanks for pointing out the issues and your valuable review.
>
> Replies to the open questions:
>
> * We have evaluated both using bert as input encoding and also without bert. Using an additional LSTM is a standard technique.
> * As evident from Table 2, DialoGPT (which utilizes GPT-2) performs worse ( BLEU: 0.76)  in the Soccer dataset compared to most models reported in Table 2. Where our model achieved a BLEU score of 1.51, which clearly shows its ability to perform better on KG-based non-goal oriented dialogues (“chit-chat”) than large pre-trained language models such as GPT-2.
> * The only modification done for the evaluation is we are using a global KG instead of a local KG as in the case of the previous work to make the task close to real-world knowledge grounded dialogues. The reason for reduced BLEU and entity f1 score is because in case of using local kg as in previous works,  the triples are unique. But in the case of global KG, the same entity and relations can have different objects attached to it and there is no way to disambiguate them. We will also mention this point in the introduction in the case of acceptance.
> * Some of the errors in the dataset occured because of the pre-processing (e.g., "predominately" got mapped to "p@objectominately") and some from the AMT experiment (where there is a question instead of an answer). This did not affect the models performance much  since the amount of errors occurred during the pre-processing is negligible.
> * The soccer dataset is more challenging since the number of factual or kg based questions are less hence all the models are not able to produce much grounded response. However, kgirnet is able to learn even from noisy datasets as well.
> * The value 1 says that the structure of the sentence is wrong in terms of grammatical errors,  malformed sentences and completeness of a sentence. where 5 denotes a perfectly formed sentence with correct grammar. For slight mistakes inside the sentence the annotator penalizes likewise and gives some other value in between 1 and 5 according to the degree of the mistake.
> * Correctness is measured by the correct answer object. reference responses are provided besides the system generated response in order to check the factual questions.
> * We have used fuzzy matching for annotating the dialogues.
> * We are using the entity f1 score based on (berant et al. -> Semantic Parsing on Freebase from Question-Answer Pairs) rather than token level used in mem2seq [18].
> * Fixed all the typos.

---

> > ### Comment · AnonReviewer1 · 2021-02-04
> > **Thank you.**
> >
> > Thank you for your comments and clarifications.

---

### Official Review · AnonReviewer2 · 2021-01-13
**Open questions remain**

**Rating:** 1
**Confidence:** 3
**Impact:** 3
**Design And Technical Quality:** 3

**Review:**

This paper proposes a novel architecture for knowledge-based dialog systems. The main idea is to embed not just the last user utterance, but also the context and the subgraph around the detected entities in order to generate a response. For this purpose, the proposed system detects (1) the target entity in the query and (2) one or multiple target relationships. These features are integrated in an end-to-end fashion into the training.

Knowledge-based dialog systems are an exciting topic, and the paper does a good job in advancing the general comprehension, importance, and performance of the field. It is great to see how knowledge bases can be used systematically to steer away from the platitudes that language-model based approaches can generate.

The proposed model works on the experimental datasets. I still have a number of questions:
- What is the purpose of the LSTM in 2.3? Is BERT not enough to encode the query, or would an LSTM alone suffice?
- How does the Laplacian help, apart from filtering our absent relations? Would it be possible to just zero out the relations in O_{dec} that do not exist for @entity? How does the 3-hop help, would a 1-hop not be sufficient?
- What is the Entity F1? I did not find its definition. Is it just the quality of the disambiguation? If so, would it be possible to use an off-the-shelf disambiguator instead? (Spotlight, AIDA,...)
- The paper euphemistically states that the proposed system is better on all accounts, except on 2 of the 6 measures. Yet, are these not exactly the important ones for the Soccer Dataset, with the Entity Score being the least relevant one?

I am also wondering why the simple baseline (where the system is trained end-to-end with place-holders for the relation names) does not work. My understanding is that the input would be “who was the director of Titanic?”, and the training target, generated by preprocessing of the original training data, would be “<r:directedBy> was the director”. Could you elaborate on why this does not work?

**Anonymity:**

Yes, I would like my review to remain anonymous.

**Reuse And Availability:**

4: High

**Strong Points:**

- interesting problem
- reasonable performance
- interesting end-to-end solution

**Subreviewer:**

I submitted this review.

**Weak Points:**

- open questions remain (see detailed reviews)

---

> ### Author Rebuttal · Authors · 2021-01-29
>
> Thanks for pointing out the issues and your valuable review.
>
> Reply to the open questions:
>
> * We have evaluated both using bert as input encoding and also without bert. Using an additional LSTM is a standard technique.
> * Yes, it is possible to zero-out the relations that are not connected to @entity. In Equation 8, adjacency matrix is used where only connected relations (with a value of 1 in the matrix) are taken into consideration for a particular @entity . Other relations will receive 0 if they are not connected to that entity. thus the system zero-out irrelevant relations. having access to more than 1-hop information, provides the system with more inference power (gives the system the ability to generate answer object which is 2 or 3 hop away from the subject or entity.
> * We use the same entity f1 definition from berant et al. (Semantic Parsing on Freebase from Question-Answer Pairs)
> * Entity-F1 scores measure how well the information from the knowledge graph is present in the system generated response with respect to the original response. Where other automatic evaluation metrics evaluate how well-matched the system generated responses with respect to the original one (usually consider word-overlap and order of the words when evaluating). Hence, Entity-F1 score is an important metric for evaluating KG-based systems where other metric focuses on computing overlap of original-predicted response pairs. So, we can conclude that all of them are relevant from their own use cases.
> * In contrast to the other e2e systems, our system tries to learn the relation rather than the answer object. which we later use to extract the answer object from the KG. This gives our system the advantage to generate better quality response (since our system already has the entity predicted, if it predicts the relation correctly then with the already obtained entity and relation information it can easily extract the answer object from the KG and generate responses which is well KG-grounded.

---

### Official Review · AnonReviewer3 · 2021-01-13
**Neural architecture that blends the semantic richness of a KG for performing answer generation**

**Confidence:** 4
**Impact:** 3
**Design And Technical Quality:** 3

**Review:**

This paper presents a QA approach based on a blend of neural architecture and a knowledge graph to produce knowledge grounded responses.

The novelty of the work is on an architecture that combines different elements already known in the literature, namely BERT, (sub)graph Laplacian embedding, and an encoder-decoder for text generation.
The architecture combines a semantic representation of the knowledge of a sub-graph that is created utilizing a Graph Laplacian encoding with the output of the encoder-decoder utilizing a Hadamard product.
The query is first transformed utilizing a BERT encoding layer to perform entity recognition (does it entity detection starting from the CLS embedding? -- re. Fig. 2 -- and an encoder-decoder architecture developed using an LSTM for synthesizing the query and computing the probabilistic distribution of the words present in the vocabulary for each step composing the word candidates.
I reckon that the numbering of the blocks in Fig. 2 is 2.X instead of 3.X, is this correct?
According to Fig.2, the step after the Hadamard product depicted as blue rumble it seems, so far if not mistaken, to be the component in charge of generating the final answer given the output of the product, however, this hasn't been properly explained (it's the only step without an in-depth description). Given the focal point it has, it's a severe lack.

The experimental setup is structured into:
- a quantitative analysis of the approach when tested with two benchmark datasets, an in-car dataset dialogues and soccer dialogues
- an ablation study for measuring the contribution of the constituting blocks
- a qualitative analysis of a few sample outputs

From the quantitative analysis, it is evident that this approach offers competitive results on both datasets with a margin improvement wrt all but KG-Copy only tested on the Soccer dialogue dataset. On the other hand, it is evident that despite the effort, the scores are still low for having an output that is perceived as acceptable: BLUE, ie word overlap, very low and likewise the entity recognition (~30% of F1). For the former, it is expected given the challenging task, for the latter it is less and it would be beneficial for the work to argument why.

The ablation focuses on investigating the contribution of both the BERT encoding and the Hadamard product. The output proves that the two are pillars of this work. This is a valuable contribution.

Qualitative results concern the human analysis of a few (how many? those in the table) samples from one (or more) evaluators. This contribution is unclear as the experimental setup is undefined thus the conclusions are questionable.

A contribution that requires further documentation/study is how the encoder-decoder is trained and if the training affects the BERT encoding. Surprisingly, the most important part of neural architecture, ie how it is trained and according to which error minimization function is briefly described in 2 lines of Sec 2.5. This is a severe lack of paper.

Some claims might require a check/revision: "KGs are an abstraction of the real world" (means?), ...

There is sudden confusion between a transformer and BERT (which utilizes a transformer architecture but not only though)

**Anonymity:**

Yes, I would like my review to remain anonymous.

**Rating:**

-1: Weak Reject

**Reuse And Availability:**

3: Medium

**Strong Points:**

- blend of a subgraph encoding using laplacian and natural language generation using an enc-dec utilizing the Hadamard product
- ablation study


**Subreviewer:**

I submitted this review.

**Weak Points:**

- lack of documentation when reporting the answer generation
- lack of details how the network is trained (objective function, learning mechanisms whether it's per-block or end-to-end,...)
- scientifically brittle qualitative evaluation

---

> ### Author Rebuttal · Authors · 2021-01-29
>
> Thanks for pointing out the issues and your valuable review.
>
> Replies to the general review:
>
> * We’ve fixed the serial number of the sections in Fig 2.
> * The blue box decides in each time step whether to fetch an answer object from the knowledge graph or to directly append the decoded word with the output. Final output is generated when it encounters the special token <EOS>
> * In quantitative analysis it happened for in_car dataset because in the in-car dataset especially in the weather domain for a particular entity-relation pair their exists multiple object (e.g. <location, day, temperature_type>, <location, day, highest_temperature>, <location, day, lowest_temperature>). Here we see that for location-day pairs there exists multiple answer objects. Hence, even for a correctly predicted entity-relation it's difficult for the model to detect the correct answer object which is a reason for the low performance in the in-car dataset despite the dataset possessing less challenge  then the Soccer dataset.
> * Human evaluation is conducted on all the data points for both the dataset. Annotators are provided with the original responses from the dataset along with the system generated responses in the order they appeared in a dialogue. We asked 3 (1 from CS and 2 from non-CS background) annotators to rate the quality of the generated responses between 1-5 with respect to the dialogue context (higher is better). The measures requested to evaluate upon a correctness (Corr.) and human-like (human) answer generation capabilities.
> * The model is trained end-to-end and the objective function is the decoder loss + the entity learning loss as mentioned in last sentence in section 2.5
> * Information of the real world can be stored in a KG in a structured (RDF triple, e.g., <subject, relation, object>) and abstract way (Paris is the capital city of France and be presented in <Paris, capital city, France>). Hence, we say KGs are an abstraction of the real world.
> * In this work, as a transformer model we use BERT, since BERT is a Bidirectional Encoder Representations from Transformers.

---

### Official Review · AnonReviewer5 · 2021-01-15
**End to End  goal and non-goal oriented dialog system with KG grounding**

**Rating:** 1
**Confidence:** 3
**Impact:** 3
**Design And Technical Quality:** 4

**Review:**

Summary
-------

In this paper, authors facilitate a dialogue system to produce knowledge grounded responses. They aim to address the challenge of integrating KGs into the dialogue generation process in an end-to-end manner. To be specific, KGs are integrated into the response generation process by learning a model which can answer using the elements of the KG. Empirical evaluation conducted on goal and non-goal oriented
dialogues has shown that proposed model perform better knowledge grounding in contrast with other state-of-the-art models.

Comments to Authors
--------------------

1. Graph Laplacian is used to encode the input sub-graph and also used for the final decoding process. Advantage is not clearly shown in the experimentation. Could also have used Graph Convolutional Networks for relation data (Schlichtkrull et al., 2018) as Table-3 shows a relation linking experiment. Comparison can be useful.
2. In Figure-2, arrows looks little bit confusing. There is no distinguishment between dotted and solid arrows.
3. <EOU> token is represented in Section-2.1. However, the token is not shown in the Figure-2 (input query).
4. Missing related work from the Generative approaches like VAE and GAN which is been extensively applied for Dialog generation.
5. Section-2.2 about entity detection which use Equation-1 to find entites is unclear. Howdoes softmax applied on hidden represetnation of BERT [CLS] output can provide entity information. It is never optimized with a loss which maximize it.


Minor Issues
----------

1. (measure via entity F1 score) --> (measured via entity F1 score)
2. Figure-2 has mismatch between section names.
3. sub-graph encoding mechanism is shown in 3 --> sub-graph encoding mechanism is shown in Figure-3
4. The code of the model is added as a supplementary material. No supplemental material is uploaded.
5. Writing the name of the dataset in Table-4 and adding more details about models will be helpful for better readability.

Overall, it is an interesting work which explores KG grounding to improve goal and non-goal oriented dialogues. Especially it is helpful to improve those domains which require external knowledge as a part of the dialog.

**Anonymity:**

Yes, I would like my review to remain anonymous.

**Reuse And Availability:**

4: High

**Strong Points:**

1. Graph Laplacian based method for encoding the input sub-graph which can elimenate KG elements.
2. End-to-End architecture which combines the KG with the dialog context.
3. Evaluation is performed on datasets which require grounded KG knowledge such as an in-car dialogue dataset and soccer dialogues for goal and non-goal oriented setting.

**Subreviewer:**

I submitted this review.

**Weak Points:**

1. Pre-trained Transformers like BERT is merely used to acquire representation of query. Looks as an over claim in the title.
2. Many fine-grain details are missing. For example, Section-2.2 about entity detection which use Equation-1 to find entites is unclear.
3. Missed several datasets like bABI Dialog (goal-oriented), DSTC2 (goal-oriented) etc what other methods such as Mem2Seq have used for comparison.

---

> ### Author Rebuttal · Authors · 2021-01-29
>
> Thanks for pointing out the issues and your valuable review.
>
> Replies for the Comments to Authors:
>
> * Given a sub-graph it captures the entity-relation pair which makes the object (answer) detection part easier. In our scenario we did it for single relation extraction by taking the maximum value from the output. It can be easily extendable by imposing threshold value in the encoded output (see Figure 3). By getting values from the encoded output above the threshold value multiple relation extraction is possible which is a clear advantage over the work of (Schlichtkrull et al., 2018).
> * It is mentioned in section 2 that "Dotted lines in the model diagram represent operations performed at a different time-step $t$ in the decoding process." In addition, we'll also add a line in case of acceptance, mentioning that the solid lines are performed once for a particular query where dotted denotes the steps performed in each time-step of the dialogue generation part.
> * We will add them in case of acceptance
> * We will add them in case of acceptance
> * The entity detection is done at sentence level, hence the CLS representation is used. As mentioned previously, we assume that a single entity is present in a question.
>
> Replies to the Minor issues:
>
> Thanks for pointing out these minor issues. We’ll fix them and add a link to the code (github repository) for the model will be added in case of acceptance.
>
> Replies to the weak points:
>
> * BERT is used for encoding the query as well as for entity prediction (C token is used form the output of BERT).
> * For entity detection we used a fully connected layer on top of CNN-based architecture. Size of the hidden layer in the fully connected part is 500 and a dropout value of 0.1 is used. We have used ReLU as the activation function. In the CNN part we have used 300 filters with kernel size 3, 4 and 5. We’ll add these details in case of acceptance.
> * DSTC-2 dataset lacks KG information for the conversations, therefore, doesn’t fit into our work since our work focuses on the KG based dialogue system. bABI is a simulated dataset while our work is focused on real-world conversations. Moreover, in bABI movie dialogue dataset is for answering factoid questions for movies and recommendations. The answers don’t contain articulate responses; the other dialogue dataset is for dialogue state tracking hence they were not suitable for the task of dialogue generation.

---

### Official Review · AnonReviewer4 · 2021-01-15
**A KG grounded dialogue system with transformer and Graph Laplacian encoder**

**Rating:** 1
**Confidence:** 4
**Impact:** 3
**Design And Technical Quality:** 3

**Review:**

# Summary

This paper presents a KG grounded dialogue system, which takes a natural language question as input and outputs the potential answer based on KG. The system first utilizes BERT embedding to encode the input sequence, then predict the entity and relation based on the sequence embedding and sub-graph embedding. Finally, the answer is generated by a sequential decoder. Generally, the proposed task is of practical value and the author gives a good solution that is tested well by extensive experiments. From my perspective, the presentation needs to be more clear. E.g., section 2.4 is confusing; the different notations without any table/figure explanation easily make readers lost.


**Anonymity:**

Yes, I would like my review to remain anonymous.

**Reuse And Availability:**

3: Medium

**Strong Points:**

Practical task and good evaluation.

**Subreviewer:**

I submitted this review.

**Weak Points:**

1. One of my major concerns is the system may only handle simple questions consists of a single node and single relation. Can this model easily be extended to some more complex questions?

2. The author claims the proposed system is capable of multi-turn dialogues and the evaluation results demonstrate that. But how to deal with the dialogue context in the model description part is unclear to me.

3. According to BLEU in Table 2, it seems this model can not generate high-quality answers in natural language, especially compared to KG-copy.

4. Efficiency, like running time, should also be evaluated and reported, since it is an important metric in a real-time dialogue system.

---

> ### Author Rebuttal · Authors · 2021-01-29
>
> Thanks for pointing out the issues and your valuable review.
>
> Replies to the Weak points:
>
> * Our proposed model can easily be extended by imposing a threshold in the output of the sub-graph encoding part. In figure 3, one can extract multiple relations from the output of the encoder which scores above the threshold value.
> the previous dialogue contexts are appended to the beginning of the dialogue generation using <SEP> token.
> * Our proposed model is built with the purpose to perform well on KG based dialogue system. The superiority in the performance in terms of knowledge grounded dialogue generation is noticeable from Table 2. Entity-F1 (KGIRNET: 34.33, KG-copy: 3.17) with a competitive BLEU score (KGIRNET: 1.51, KG-copy: 1.93)
> * We’ll add the run times in case of acceptance.

---

### Decision · Program_Chairs · 2021-02-23

**Decision:**

Accept with shepherding

**Comment:**

This meta review summarizes the strengths and weaknesses pointed out by the reviewers. There was consensus among the reviewers that the paper addresses an interesting and important problem within scope of the conference. The reviewers also agreed that the proposed end-to-end approach (using a clever combination of BERT and subgraph encoding via the Graph Laplacian) is sensible and constitutes a solid technical contribution. However, as evident from the initial reviews, the reviewers also pointed out some issues that need addressing, which the authors already acknowledged in their rebuttal. For the paper to be accepted, the authors are kindly asked to address the following issues and submit a revised version of the paper:

[Task 1] Extend the experimental evaluation by runtimes for all approaches considered (cf. Review 4).

[Task 2] Clarify the applicability of the proposed approach for complex questions -- ideally by means of an illustrating example (cf. Reviews 4 and 5).

[Task 3] Include a detailed description of the human evaluation via Amazon Mechanical Turk (cf. Review 3).

[Task 4] Address reviewers' comments regarding small presentation issues, missing technical details (e.g., definition of Entity F1-Score), and relevant related work (cf. Review 5).

Please submit the revised version of the paper by March 17th 2021. Should you have any questions or want feedback on your changes, feel free to contact Klaus Berberich (klaus.berberich@htwsaar.de).

[Task 5] Include a link to the source code (e.g., in a GitHub repository).